# Blood–brain barrier opening in Alzheimer's disease using MR-guided focused ultrasound

Nir Lipsman[1,2,3], Ying Meng[1,2], Allison J. Bethune[2,3], Yuexi Huang[4],
Benjamin Lam[2,5], Mario Masellis[2,5], Nathan Herrmann[2,6], Chinthaka Heyn[4,7],
Isabelle Aubert[2,4,8], Alexandre Boutet[7], Gwenn S. Smith[9],
Kullervo Hynynen[4,10,11] & Sandra E. Black[2,5]

Magnetic resonance-guided focused ultrasound in combination with intravenously injected microbubbles has been shown to transiently open the blood–brain barrier, and reduce beta-amyloid and tau pathology in animal models of Alzheimer's disease. Here, we used focused ultrasound to open the blood–brain barrier in five patients with early to moderate Alzheimer's disease in a phase I safety trial. In all patients, the blood–brain barrier within the target volume was safely, reversibly, and repeatedly opened. Opening the blood–brain barrier did not result in serious clinical or radiographic adverse events, as well as no clinically significant worsening on cognitive scores at three months compared to baseline. Beta-amyloid levels were measured before treatment using [$^{18}$F]-florbetaben PET to confirm amyloid deposition at the target site. Exploratory analysis suggested no group-wise changes in amyloid post-sonication. The results of this safety and feasibility study support the continued investigation of focused ultrasound as a potential novel treatment and delivery strategy for patients with Alzheimer's disease.

[1] Division of Neurosurgery, Sunnybrook Health Sciences Centre, University of Toronto, Toronto M4N 3M5, Canada. [2] Hurvitz Brain Sciences Research Program, Sunnybrook Research Institute, Sunnybrook Health Sciences Centre, University of Toronto, Toronto M4N 3M5, Canada. [3] Harquail Centre for Neuromodulation, Sunnybrook Research Institute, Toronto M4N 3M5, Canada. [4] Sunnybrook Research Institute, Sunnybrook Health Sciences Centre, University of Toronto, Toronto M4N 3M5, Canada. [5] Division of Neurology, Sunnybrook Health Sciences Centre, University of Toronto, Toronto M4N 3M5, Canada. [6] Division of Geriatric Psychiatry, Sunnybrook Health Sciences Centre, University of Toronto, Toronto M4N 3M5, Canada. [7] Department of Medical Imaging, Sunnybrook Health Sciences Centre, University of Toronto, Toronto M4N 3M5, Canada. [8] Department of Laboratory Medicine and Pathobiology, University of Toronto, Toronto M5S 3H7, Canada. [9] Department of Psychiatry and Behavioral Sciences and Radiology and Radiological Sciences, Johns Hopkins University, Baltimore, MD 21218, USA. [10] Department of Medical Biophysics, University of Toronto, Toronto M5S 3H7, Canada. [11] Institute of Biomaterials and Biomedical Engineering, University of Toronto, Toronto M5S 3H7, Canada. Correspondence and requests for materials should be addressed to N.L. (email: nir.lipsman@sunnybrook.ca)

Alzheimer's disease (AD) is the most common neurodegenerative disorder, marked by progressive decline in memory and cognition over decades. The pathologic hallmarks of AD include extracellular Aβ plaques and intracellular neurofibrillary tangles, changes that contribute to widespread metabolic and neurochemical disturbances, and which culminate in neuronal degeneration and cell death. For the last three decades, the amyloid cascade hypothesis, which posits Aβ deposition as a key initial step in the pathogenesis of AD, has been the dominant theory driving treatment development[1]. While the role of amyloid burden in the development and progression of clinical symptoms remains unclear and under active investigation, plaque clearance has been a key target of numerous clinical trials. Results to date have largely been disappointing[2,3], although one recent trial linked significant reduction in amyloid to stabilization of cognitive decline at 1 year[4].

The blood–brain barrier (BBB) is a major obstacle for the effective delivery of therapeutic compounds to the brain, imposing size and biochemical restrictions on the passage of molecules[5]. Various strategies to overcome the BBB have been investigated, including direct intracranial infusion[6], convection enhanced delivery[7], diuretic agents[8], and biomimetics[9]. These approaches have been limited by lack of specificity, safety concerns, and a failure to achieve adequate concentrations of delivered compounds to sufficient volumes of brain tissue[10,11]. A safe and effective means of bypassing the barrier temporarily could aid in delivering even large molecules, such as antibodies and growth factors, directly to brain pathology.

Magnetic resonance-guided focused ultrasound (MRgFUS) is an emerging non-invasive surgical modality that, when coupled with injected microbubbles, transiently opens the BBB with a high degree of spatial and temporal specificity. Comprised of a phased array transducer system of 1024 individually steered elements, the current clinical MRgFUS device can target brain regions with sub-millimetric accuracy, using real-time feedback for monitoring and intraoperative image guidance[12]. At high frequencies, focused ultrasound has been used to non-invasively ablate the ventro-intermediate nucleus of the thalamus in patients with refractory essential tremor, with a safety and efficacy profile comparable to open neurosurgical approaches[13,14]. Other clinical trials have investigated thermoablative applications in tremor due to Parkinson's disease[15] and obsessive compulsive disorder[16]. At lower frequencies, the interaction of ultrasound with injected microbubbles results in transient disruption of the BBB[17].

Focused ultrasound has been used in animal models of diseases including brain tumors, Parkinson's disease, and AD, wherein the delivery of a range of therapeutic substrates has been enhanced, including antibodies[18–20], chemotherapy[21], nanoparticles[22], Herceptin[23], viruses[24], and stem cells[25]. In transgenic mouse models of AD, ultrasound was used to deliver antibodies against beta-amyloid and tau, with significant reductions in pathology and a positive impact on memory performance[18,19]. Further studies in mouse models of amyloidosis demonstrated that even without exogenous antibody administration, BBB disruption by focused ultrasound reduced plaque burden, triggered neuronal plasticity, and prevented spatial memory deficits[26–28].

Given the compelling preclinical evidence, we investigate for the first time the use of non-invasive MRgFUS to open the BBB in human patients, with mild-to-moderate, amyloid-positive AD. Our primary aim is to evaluate the clinical safety and technical feasibility of this procedure, and secondarily to measure the influence, if any, on clinical and beta-amyloid imaging markers of AD. We find that the BBB can be safely, temporarily, and repeatedly opened in an amyloid-rich brain region with a high degree of anatomic specificity.

## Results

**Study patients**. We enrolled five patients, three men and two women, with a mean age of 66.2 years (Fig. 1, Table 1). Average baseline Mini-Mental State Examination (MMSE) was 22.8, suggestive of mild-to-moderate stage disease. The level of amyloid on [18F]-florbetaben positron emission tomography (PET) exceeded the cutoff for amyloid positivity (standardized uptake value ratio; SUVr ≥1.43) for all patients[29]. Four patients completed both first and second stages of sonication. Patient 4 developed a respiratory illness unrelated to the procedure shortly before stage 2, and we elected not to proceed with the second stage given the unknown risk of opening the BBB in the context of an active infection.

**Primary outcome**. The BBB was successfully opened in all patients who underwent the focused ultrasound procedure (Fig. 2 and Supplementary Figure 1). We primarily targeted white matter in the frontal lobe, attempting to be specific to the dorsolateral prefrontal cortex where possible, given anatomic constraints. The average maximum sonication power was 4.6 W with an average of 3.6 sonications administered for stage 1 and 4.5 W for 7.5 sonications for stage 2 (Supplementary Table 1). BBB opening was achieved predictably at approximately 50% power at which cavitation was observed during a ramp test. Immediately after sonication, a discrete rectangular-shaped gadolinium enhancement can be seen in the targeted region on T1-weighted images (Fig. 2). At 24 h following the procedure, there was resolution of enhancement in the targeted region, indicating closure of the BBB (Fig. 2).

No patient experienced a serious adverse event during this study. There were no deaths, hemorrhages, swelling, or neurologic deficits on the day of procedure or during follow-up. One patient showed a transient increase in the Neuropsychiatric Inventory—Questionnaire (NPI-Q) score, during the 1-month visit following stage 2 (Table 2). All patients were discharged the morning after their procedures. Patient 1 experienced headaches during follow-up, which resolved with updating his prescription for his vision.

Radiologically, there was no evidence of intracerebral hemorrhage or swelling. In two patients, patient 1 and 5, discrete round hypointensities on gradient echo were seen immediately after sonication (Supplementary Figure 2). Although these findings may indicate microhemorrhage, they resolved by the 24-h follow-up magnetic resonance imaging (MRI), rather than persisting, as is more typical of hemosiderin-type staining from microbleeds.

**Secondary outcome**. For our secondary outcome, we did not detect a clinically significant change between 3 months and baseline, on tests of patient cognition or daily functioning. Table 2 lists the clinical psychometric data at baseline and follow-up visits. Group PET changes in the regions of interest (ROIs) after stages 1 and 2 compared to baseline were −0.14 ± 0.22 (standard deviation, $n = 5$) and −0.08 ± 0.21 ($n = 4$), respectively, which were not statistically significant ($p > 0.2$, paired $t$-test, Fig. 3, Supplementary Fig. 3–7, Supplementary Tables 2, 3).

## Discussion

We demonstrate for the first time, safe, reversible, and repeated, non-invasive opening of the BBB using MRgFUS in patients with amyloid-positive AD. BBB opening was achieved with less than 1% of the energy required to create a thermocoagulative lesion, thereby enhancing the safety and expanding the treatment envelope of the procedure[30]. We observed no serious clinical or radiographic adverse events, with progressively larger brain regions opened, and complete closure of the barrier within 24 h.

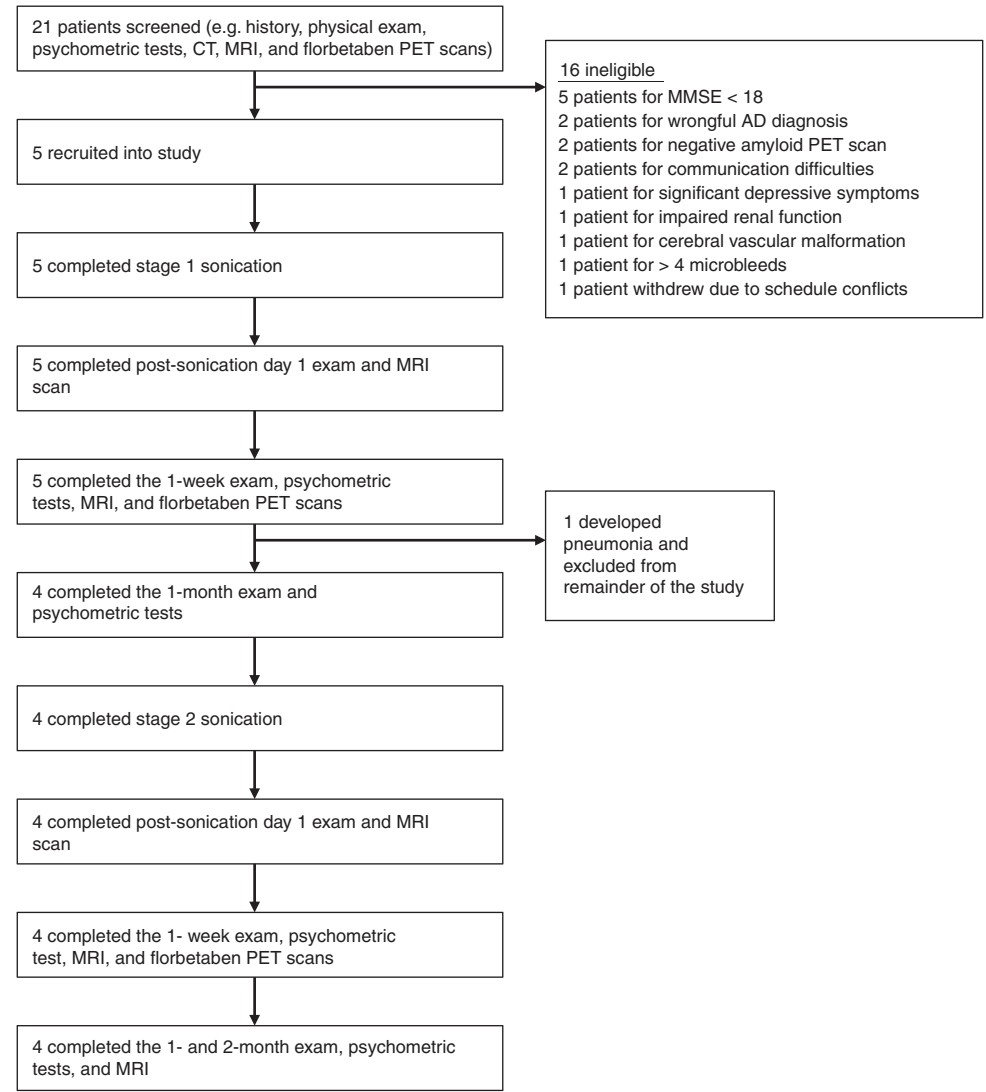

**Fig. 1** Overview of the study. Flow chart illustrates the study design and overview of patients screened and enrolled in the study

**Table 1 Patient demographics and baseline clinical characteristics**

| | Sex | Age | Duration of illness (years) | Baseline MMSE | Family history of AD | Comorbidities | Medications |
|---|---|---|---|---|---|---|---|
| Patient 1 | Male | 64 | 6 | 20 | No | Depressive symptoms, dyslipidemia, chronic obstructive pulmonary disease | Donepezil, rosuvastatin, sertraline |
| Patient 2 | Male | 64 | 2 | 25 | No | Depressive symptoms, hypertension, dyslipidemia, mild obstructive sleep apnea | Donepezil, rosuvastatin, escitalopram |
| Patient 3 | Female | 63 | ¼ | 22 | No | Depression, dyslipidemia, asthma | Donepezil, budesonide/formoterol, atorvastatin |
| Patient 4 | Male | 78 | 3 | 25 | Dementia | Depressive symptoms | Omeprazole, meloxicam, escitalopram, donepezil |
| Patient 5 | Female | 62 | 4 | 21 | Dementia | Depressive symptoms, hypertension, hyperlipidemia, inflammatory bowel disease, arthritis | Proranolol, perindopril, meloxicam, duloxetine, lansoprazole, atorvastatin, zopiclone, indapamide, donepezil |
| Mean (SD) | | 66.2 (6.6) | 3.1 (2.2) | 22.6 (2.3) | | | |

*SD* standard deviation

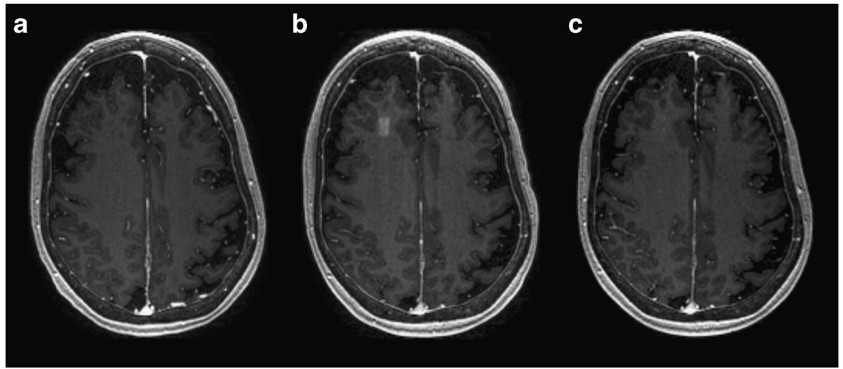

**Fig. 2** MRI demonstration of blood–brain barrier opening and closure. Axial T1-weighted gadolinium MR images of patient 5 at **a** baseline, **b** immediately after stage 2 sonication and blood–brain barrier (BBB) opening, and **c** at 24 h after procedure. Contrast extravasation within the $10 \times 10 \times 7$ mm$^3$ sonicated volume in the right frontal lobe is seen immediately after the procedure, demonstrating increased BBB permeability. At 24 h after the procedure, there is no significant extravasation of contrast in the area, suggesting BBB closure

### Table 2 Psychometric measures

|  | Baseline | Stage 1 | | Stage 2 | | |
|---|---|---|---|---|---|---|
|  |  | 1 week | 1 month | 1 week | 1 month | 2 months |
| **MMSE** |  |  |  |  |  |  |
| Patient 1 | 20 | 22 | 21 | 22 | 23 | 22 |
| Patient 2 | 25 | 23 | 20 | 26 | 29 | 23 |
| Patient 3 | 22 | 22 | 19 | 20 | 23 | 18 |
| Patient 4 | 25 | 21 | – | – | – | – |
| Patient 5 | 21 | 23 | 22 | 25 | 25 | 24 |
| Mean (SD) | 22.6 (2.3) | 22.2 (0.8) | 20.5 (1.3) | 23.3 (2.8) | 25.0 (2.8) | 21.8 (2.6) |
| **ADAS-cog** |  |  |  |  |  |  |
| Patient 1 | 19 | 21 | 17 | 22 | 18 | 19 |
| Patient 2 | 21 | 16 | 23 | 15 | 12 | 22 |
| Patient 3 | 30 | 29 | 32 | 28 | 32 | 25 |
| Patient 4 | 19 | 15 | – | – | – | – |
| Patient 5 | 19 | 20 | 17 | 15 | 20 | 25 |
| Mean (SD) | 21.6 (4.8) | 20.2 (5.5) | 22.3 (7.1) | 20.0 (6.3) | 20.5 (8.4) | 22.8 (2.9) |
| **GDS** |  |  |  |  |  |  |
| Patient 1 | 0 | 0 | 1 | 2 | 2 | 2 |
| Patient 2 | 4 | 2 | 4 | 5 | 2 | 2 |
| Patient 3 | 1 | 1 | 1 | 1 | 1 | 1 |
| Patient 4 | 2 | 3 | – | – | – | – |
| Patient 5 | 4 | 2 | 4 | 3 | 2 | 3 |
| Mean (SD) | 2.2 (1.8) | 1.6 (1.1) | 2.5 (1.7) | 2.8 (1.7) | 1.8 (0.5) | 2.0 (0.8) |
| **NPI-Q** |  |  |  |  |  |  |
| Patient 1 | 0 | 0 | 0 | 2 | 27 | 6 |
| Patient 2 | 0 | 0 | 0 | 0 | 0 | 16 |
| Patient 3 | 0 | 1 | 1 | 0 | 1 | 2 |
| Patient 4 | 0 | 3 | – | – | – | – |
| Patient 5 | 2 | 1 | 0 | 0 | 0 | 0 |
| Mean (SD) | 0.4 (0.9) | 1.0 (1.2) | 0.3 (0.5) | 0.5 (1.0) | 7.0 (13.3) | 6.0 (7.1) |
| **ADCS** |  |  |  |  |  |  |
| Patient 1 | 63 | 64 | 62 | 65 | 66 | 68 |
| Patient 2 | 64 | 65 | 61 | 64 | 62 | 49 |
| Patient 3 | 67 | 62 | 66 | 69 | 58 | 66 |
| Patient 4 | 72 | 66 | – | – | – | – |
| Patient 5 | 76 | 72 | 75 | 72 | 75 | 70 |
| Mean (SD) | 68.4 (5.5) | 65.8 (3.8) | 66.0 (6.4) | 67.5 (3.7) | 65.3 (7.3) | 63.3 (9.6) |

*MMSE* Mini-Mental State Examination, *ADAS-cog* Alzheimer's Disease Assessment Scale—cognitive, *ADCS-ADL* Alzheimer's Disease Cooperative Study Group—Activities of Daily Living, *GDS* Geriatric Depression Scale, *NPI-Q* Neuropsychiatric Inventory Questionnaire

No previous studies have reported non-invasive and reversible BBB opening in humans. Previously published work investigated pulsed ultrasound via a surgically implanted device, in patients with malignant brain tumors[31]. Our trial utilized a non-invasive device, guided by real-time imaging and thermometry, demonstrating BBB opening with millimeter accuracy, and its successful closure within 24 h.

In this pilot study, we opened the BBB twice in the right frontal lobe, 1 month apart with the second volume twice the first. Because brains affected by AD are often atrophic, it was necessary

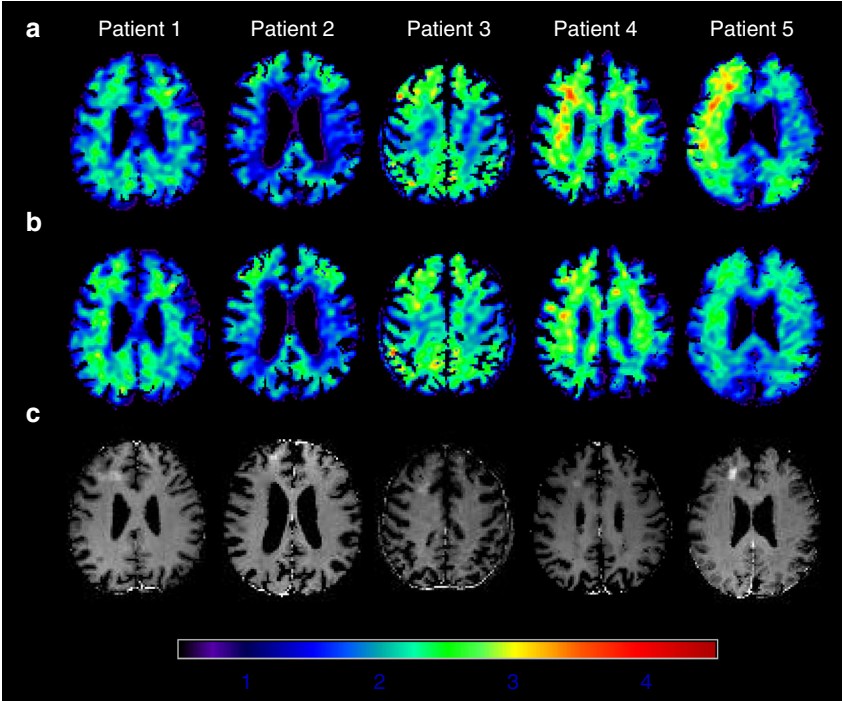

**Fig. 3** [18F]-Florbetaben uptake on PET before and after treatment. Standardized uptake value ratio images (SUVr) in the corresponding axial planes for [18F]-Florbetaben PET scans at **a** baseline and **b** approximately 1 week after sonication. **c** Gadolinium extravasation on T1-weighted MR images immediately after the blood–brain barrier disruption procedure demonstrates the targeted region

---

**Table 3 Inclusion and exclusion criteria**

*Inclusion criteria*
Male or female between age of 50–85
Probable AD consistent with NIA/AA criteria
Modified Hachinski Ischemia Scale ≤4
Mini Mental State Exam 18–28
Short form Geriatric Depression Scale ≤6
If being treated with AChEI and/or Memantine, has been on medication for ≥4 months with a stable dose for ≥3 months
Beta-amyloid deposition on [18F]-florbetaben PET in the right frontal lobe
ASA physical status classification I-III

*Exclusion criteria*
Contraindications to MRI, MRI contrast, or ultrasound contrast
MRI findings of active or acute neurological process (e.g. infection, tumor) or macrohemorrhage or >4 lobar microbleeds
≥30% of the skull area traversed by sonication is covered by scars, scalp disorder or atrophic scalp
Significant cardiac disease
Uncontrolled hypertension
Predisposition for bleeding
Known cerebral or systemic vasculopathy
Frequency or severity of ≥2 on Delusion, Hallucination, or Agitation/Aggression subscales of the NPI-Q
Impaired renal function
Severe chronic respiratory disorders

*NIA/AA* National Institute on Aging/Alzheimer's Association, *AChEI* acetyl-cholinesterase inhibitor, *ASA* American Society of Anesthesiologists

---

to adjust the axial dimensions of the planned target so as to avoid sulci and blood vessels. Doing so underscored the importance of in vivo imaging during these procedures, and the reliance on the clinical and imaging feedback afforded by the ultrasound system used for this trial. Acoustic feedback monitoring, contrast enhanced MR images, and physical examinations between sonications were key in determining a safe power for BBB opening and detecting any adverse events such as bleeding. In general, the procedure was well tolerated by all patients.

Efficient MRgFUS-mediated BBB opening is influenced by a number of procedural and technical variables. Determining the optimal power via a ramp test is the first step to achieving uniform BBB opening. Other factors include microbubble size and dose, volume and type of tissue targeted, microbubble handling, and coordination between sonication and microbubble injection. As with any procedure, additional experience will enhance its efficiency and success, and while we achieved successful BBB opening in all patients, results were most uniform in the last two patients (Supplementary Fig. 1).

Exploratory analysis of the [18F]-florbetaben PET results did not identify a clear effect, in either direction, of MRgFUS BBB opening on beta-amyloid deposition. The BBB is important in maintaining brain homeostasis, and its role in the clearance of pathologic proteins is complex. While the integrity of the BBB in AD is controversial[32,33], preclinical studies have shown improvements in both pathology and phenotype following focused ultrasound, providing evidence that BBB opening alone, or in conjunction with a therapeutic, could be a potential treatment for patients. The mechanisms of beta-amyloid clearance following focused ultrasound remain under investigation. MRgFUS BBB opening in mouse models has been shown to allow the entry of endogenous antibodies and blood-borne proteins in targeted areas of the brain, which could contribute to beta-amyloid opsonization and clearance by glial cells[26,27]. In addition, MRgFUS BBB opening may increase the permeability of the glymphatic system as fluorescent-tagged Aβ protein clearance follows along the perivascular spaces of arterioles and venules[34].

This study has several important limitations. First, our sample size is small, and although we provide evidence of clinical and radiographic safety, this limits generalizability. Second, the study was not designed to study efficacy, and we cannot draw conclusions about what effect, if any, focused ultrasound might have on the clinical symptoms of AD or on beta-amyloid clearance or deposition. Four enrolled patients were further under the age of 65, suggesting the possibility of earlier onset disease, distinct from sporadic AD. The relatively young age of enrolled patients may further affect generalizability given that prevalence of AD increases with age. As this was a phase I, pilot study, our objective was to obtain safety data in demonstrated amyloid-positive AD, with subsequent larger trials, permitting more detailed subgroup analyses.

For this study, we targeted a non-eloquent brain region, and did not combine focused ultrasound with a therapeutic. Instead, we chose to focus entirely on the question of safety and feasibility, and whether in the context of known amyloid pathology, non-invasive BBB opening can be done reversibly and repeatedly in progressively larger volumes. Focused ultrasound, when combined with injected microbubbles, opened the BBB within seconds, and with a high degree of specificity and accuracy, suggesting the possibility of targeting much larger volumes. Furthermore, the control over the spatial location of the target area, afforded by real-time imaging with sub-millimeter resolution, suggests that the BBB may be opened in areas with complex anatomy such as the hippocampus and other eloquent cortical and subcortical structures. Future studies targeting such structures at earlier stages of the illness, and in a larger group of patients, will help determine whether focused ultrasound BBB opening alone, or with a therapeutic, can be of any benefit for this devastating disease.

## Methods

**Study design and participants.** This open label, prospective, proof-of-concept, phase I trial was designed to study the safety and feasibility of repeated BBB opening in patients with AD with demonstrated amyloid deposition in the targeted area. To improve safety, the study was divided into two stages, graded by the volume of brain tissue for BBB opening. Moreover, presumed non-eloquent cortex in the right frontal lobe, namely the superior frontal gyrus white matter of the dorsolateral prefrontal cortex (DLPFC), was selected to minimize potential complications in the event of bleeding or mass effect from vasogenic edema. The study was approved by the Research Ethics Board at Sunnybrook Health Sciences Centre (SHSC) and Health Canada. This study was registered with ClinicalTrial.gov number NCT02986932, and Health Canada number 195168. Prior to enrollment, all patients and their primary caregivers provided informed consent to the study, and publication of radiologic images.

Figure 1 outlines the study design. Patients between age 50 and 85 with mild-to-moderate AD[35] with an MMSE score equal to or greater than 18 were eligible for the study. Detailed inclusion and exclusion criteria are listed in Table 3. In general, patients were referred to the study by neurologists and geriatricians. They were

excluded if they had any contraindications to MRI, gadolinium or ultrasound contrast (Definity®), increased risk of bleeding, active intracranial diseases such as brain tumors or vascular malformations, or significant cardiovascular, pulmonary, and renal disease. During screening, patients underwent confirmation of their diagnosis by an expert in cognitive neurology, a pre-surgical anesthetic evaluation, baseline psychometric tests, and radiographic investigations with CT, MRI, and [18F]-florbetaben PET CT scans. Twenty-one patients were screened, of which five entered and completed the study, between March and September 2017.

**MRgFUS procedure.** We used a focused ultrasound device consisting of 1024 individual transducers with a frequency of 220 kHz (ExAblate Neuro; InSightec Haifa). The device integrates intraoperative imaging, which was used for interim evaluations of the patient, and real-time acoustic monitoring to support decision-making on sonication parameters. On the day of the procedure, a Cosman-Roberts-Wells (CRW) stereotactic frame was fixed to the patient's head under local anesthetic. The frame was then coupled to the helmet transducer array, with the patient entering the MRI supine and awake. A safety switch was given to the patient to abort the procedure in case of discomfort or pain. The patient was examined and questioned for adverse events after each sonication.

A 3-Tesla MRI (Signa MR750; GE Healthcare, Milwaukee, Wis.) was used to obtain T1, T2 (fast spin echo), and T2* (gradient echo) weighted images for surgical planning. A region in the right frontal lobe was then selected for BBB opening. To minimize the risk, we avoided areas containing sulci and vessels within two contiguous MRI slices in each of the axial, sagittal, and coronal planes. Once the target region was identified, patients received a weight-based intravenous injection of microbubble contrast (Definity®) (4 μl/kg), followed shortly by the application of low-frequency focused ultrasound to the target. MR thermometry was used to monitor tissue temperature at the sonicated region in real time. The sonication parameters were limited by the clinical device hardware and software, and corresponded to those previously tested in large animal models[36].

At each new target, a power ramp test was performed with the first microbubble injection. This test involves applying short sonications with increasing power in 5% increments until the device hydrophones detect a sub-harmonic acoustic feedback from the target, indicating a cavitation. Subsequent sonications are then performed at 50% of this 'cavitation threshold' power. The ramp test was developed from preclinical studies to determine the optimal power required for safe opening of the BBB[36,37]. Sonication volumes covered a rectangular spot approximately 9 mm by 9 mm, comprised of 3-by-3 grid of spots, each 3 mm in diameter. For the last three patients, given the extent of atrophy on their MRI, a 2-by-2 grid was utilized, yielding a spot approximately 5 mm by 5 mm. The device electronically steered the ultrasound through each grid for 50 s total, sonicating each spot with 2 ms on and 28 ms off bursts for 300 ms, with a repetition interval of 2.7 s (duty cycle 0.74%). For stage 2, performed approximately 1 month following stage 1, the procedure was repeated, opening the BBB at the original location as well as at an adjacent area, following the same protocol, but doubling the volume of tissue opened.

After completion of the sonication protocol, a gadolinium-enhanced T1 sequence was performed to verify definitive evidence of BBB opening. Contrast enhancement at the targeted region signified the end of the procedure. The patient was then taken out of the scanner, the stereotactic frame removed, and additional high-resolution MRI sequences obtained. Patients were admitted to the surgical short stay unit for overnight observation.

**Outcomes.** The primary outcomes were clinical and radiographic safety as well as technical feasibility of reversible and repeated BBB opening. Successful barrier opening and restoration was determined, respectively, by gadolinium leakage immediately after sonication and by the absence of enhancement 1 day after sonication at the target region on T1-weighted contrast images. Safety was measured by clinical exam during the procedure and at each follow-up, as well as radiographic examination for any adverse events, including hemorrhage, swelling, or mass effect. Follow-up visits were scheduled for 1 day, 1 week, and 1 month after each procedure, as well as 2 months following the second procedure (Fig. 1). Adverse events were recorded and monitored in a prospective fashion.

Secondary outcomes were Alzheimer's-specific psychometrics, and exploratory outcomes included regional changes in [18F]-florbetaben binding on PET. Psychometric tests administered at 1-day, 1-week, 1-month, and 2-months after sonication included the MMSE, Geriatric Depression Scale (GDS), Alzheimer's Disease Assessment Scale-cognitive (ADAS-Cog), NPI-Q, and Alzheimer's Disease Cooperative Study Activity of Daily Living Scale (ADCS-ADL).

**[18F]-Florbetaben PET CT image acquisition and analysis.** PET CT scans to measure beta-amyloid deposition were performed at baseline and 1 week following each procedure. A transmission scan followed by a 20-min emission scan (four frame/5 min each) were acquired on the Phillips Gemini PET CT (3D mode) starting at 90 min after an 8 mCi ± 20% (n = 14) radiotracer injection of [18F]-florbetaben. The SUVr was calculated on a voxel-wise basis by dividing the summed PET images by the cerebellar gray matter ROI, consistent with other [18F]-florbetaben studies[29]. To derive the cerebellar gray matter ROI, the T1-weighted MR images were processed with the Freesurfer pipeline (version 5.1; http://surfer. nmr.mgh.harvard.edu/). The radioactivity in the cerebellar reference region was extracted after mapping the cerebellar gray matter ROI to the co-registered

[$^{18}$F]-florbetaben scans. Image preprocessing was performed with statistical parametric mapping, version eight (SPM8, Institute of Neurology, London). PET-to-PET and MR-to-PET registrations were performed using the normalized mutual information algorithm, and images were spatially normalized into standard 3D space relative to the anterior commissure using the Alzheimer's Disease Neuroimaging Initiative (ADNI) template[38].

To measure the effects of MRgFUS-mediated BBB opening on beta-amyloid deposition, the sonication ROIs were manually delineated as the contrast enhanced areas plus adjacent gray and white matter on gadolinium MR images over 6–8 contiguous slices. These ROIs were mapped onto the co-registered T1-weighted MR and PET scans. Of note, the ROIs for stage 2 were larger than stage 1, consistent with the larger sonication volumes.

**Role of funding source**. InSightec, the manufacturer of the ExAblate device used in this study, was the regulatory sponsor and had no role in study design, data collection, analysis, or interpretation. This study was funded by a grant from the Focused Ultrasound Foundation, a non-profit organization that funds research into clinical applications of ultrasound. The corresponding author had full access to all the data in the study and had final responsibility for the decision to submit for publication.

**Data availability**. The data generated during the current study are available from the corresponding author on request.

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

## Acknowledgements

This study was supported by the Focused Ultrasound Foundation and Sunnybrook Foundation. The authors would like to thank Ruby Endre and Garry Detzler for their technical support in the study, and patients and their families for their involvement and contributions. N.L. is grateful for the philanthropic gifts to the Sunnybrook Foundation, Sunnybrook Research Institute, and Harquail Centre for Neuromodulation. S.E.B. is grateful for the financial support from the Sunnybrook Research Institute, the Brill Chair, Department of Medicine at University of Toronto, Sunnybrook Foundation, and the Toronto Dementia Research Alliance. M.M. acknowledges financial support from the Department of Medicine at SHSC, University of Toronto, and Sunnybrook Foundation. N.H. acknowledges financial support from the Lewar Chair, and Department of Psychiatry at SHSC and University of Toronto. B.L. was supported by the LC Campbell Foundation and the Slaight Foundation.

## Author contributions

N.L., Y.M., Y.H., M.M., N.H., I.A., K.H., and S.E.B. planned and designed the study. N.L., Y.M., A.J.B., Y.H., B.L., C.H., K.H., and S.E.B. conducted the study. C.H., A.B., and G.S. analyzed the imaging data. B.L., S.E.B., M.M., and N.H. screened the patients, and

interpreted cognitive outcome data. N.L., Y.M., Y.H., and K.H. performed the procedures. N.L., Y.M., A.J.B., B.L., K.H., and S.E.B. contributed to the analysis and interpretation of clinical and safety data. N.L. and Y.M. wrote the first manuscript draft, which was critically reviewed and revised by all authors.

## Additional information

**Competing interests:** N.L. (chair), K.H., and S.E.B. have received an honorarium for serving on an expert steering committee on focused ultrasound in AD. K.H. is an inventor on intellectual property owned by Brigham and Women's hospital in Boston and Sunnybrook Research Institute in Toronto related to intracranial focused ultrasound technology. Y.M., A.J.B., Y.H., B.L., M.M., N.H., C.H., I.A., A.B., and G.S.S. declare no competing interests.

