## [Peer Review File · Nature Communications]

Reviewers' comments:

Reviewer #1 (Remarks to the Author):

Lipsman and colleagues pursue an innovative approach to treat Alzheimer's disease that so far has remained incurable. Their paper reports a small safety study with five MCI patients that had undergone a targeted opening of the blood-brain barrier (BBB) in the non-eloquent cortex in the right frontal lobe using a two-staged approach, with a targeted area that was twice as large in stage 2 compared to that in stage 1. To achieve BBB opening the team of Dr Lipsman and colleagues used MR-guided focused ultrasound. The team reports safety and successful BBB opening as the primary outcome) and a possible reduction in amyloid-beta as measured with [18F]-florbetaben PET at the targeted region in some patients, suggesting an effect on plaque clearance (secondary outcome)..

Safety: From a safety point of view this proof-of-concept study is promising as proven by a battery of read-outs (radiographic, neuropsychological).

(1) Could the authors expand why they avoided areas containing sulci and vessels. Do they expect an effect of the microbubbles on large vessels?

BBB opening: It is good to find that the BBB closes after one day as shown by gadolinium leakage. The authors report about a learning curve and indeed, as Fig S1 shows, BBB opening has not been uniformly achieved.

(2) How did the team space out the sonications? Did they do stage 1 and then 2 on patient 1, and then move to patient 2, then 3, then 4 and 5, or did they do all stage 1 sonications pretty much at the same time?

(3) Lines 133 to 138 require some further information. What do the authors mean by 3-by-3 spot grid -2-by-2 for the last three patients? How many spots in total? Also, what was done with the first two patients? By sonicating each spot in burst mode for 300ms how do they end up being 'precisely 2ms on and 28ms off'?

Amyloid:

(4) An 18F-flobetaben PET scan is shown for patient 5 (Fig 3). Why is there a hotspot of PET signal in the right hemisphere? Considering the very small area that has been targeted how do the authors explain the dramatic effect in the entire hemisphere? Is it possible that in Fig 3C less label has been taken up?

(5) How do the scans look like for patients 1-4?

(6) Looking at Table S2, it seems that for the first two patients the SUVr value even goes up. The values were calculated by normalizing to the cerebellar gray matter. How does the values look like for the corresponding frontal area on the contralateral non-targeted side (again by normalizing for the cerebellar gray matter).

Typo line 131/132: 'Further sonications were delivered energy at half the power ...

In my view the safety and BBB opening is exciting and convincing, but the reduction in amyloid is based on one patient only (with a surprising asymmetry) and should be removed as a statement from both the abstract and the body of the text.

Reviewer #2 (Remarks to the Author):

Comments for Author:

Lipsman et al describe opening the blood brain barrier in patients with Alzheimer's disease using focused ultrasound and microbubbles. They suggest that using their methodology it was possible to demonstrate a transient opening that occurred following treatment, in the blood brain barrier that appeared to have some association with reduction in fibrillary amyloid in a focused area of the brain in 5 subjects with Alzheimer's disease, but due to small numbers and lack of placebo control, it is unclear whether there were any effects on cognition.

The results are interesting from a treatment standpoint as well as documenting potential safety prior to possible further steps towards a therapeutic clinical trial.

Comments as follows:

i) Patients selected appear to be relatively young, at least 4 of the 5 were in their 60s; raising the question was there a bias toward accepting younger patients for some reason? More detail regarding why patients were excluded would be helpful.

ii) What is the "ramp" test?

iii) Several of the patients were on SSRIs that have been suggested to utilize a transporter protein to cross the blood brain barrier. Any comments on effects or interaction of their technique with these drugs.

iv) It was stated technical performance improved with every patient and results improved with each following patient. This does not describe study design and conduct in a reproducible way for the reader. What aspects of methodology improved?

v) Florbetapir was used before these studies. With application of technique were the regions treated all previously documented positive for amyloid? Were there any broader effects on reduction in brain amyloid?

vi)

In summary very interesting data, but report would be improved with more details regarding methodology and study design.

Reviewer #3 (Remarks to the Author):

Lipsman and colleagues studied the safety of open treatment with MR-guided ultrasound in the right frontal lobe with intravenously injected microbubbles performed twice over 4 weeks in 5 patients with mild-moderate Alzheimer's disease. They hypothesized that this technique would alter the blood brain barrier. They used transient local gadolinium enhancement as evidence of change in the blood brain barrier. They state that the procedure was well-tolerated but there were some safety signals.

Abstract-The statement regarding beta amyloid reduction is overstated and should be removed. The statement regarding safety should be revised.

Methods-Please report the reasons that 16 participants were excluded.

- 4/5 participants were <65 which is not representative of the age range of individuals with sporadic AD. Testing the safety and efficacy of this technique in early onset AD may not be generalizable to older people with sporadic AD as early onset patients have a higher APOE carriage rate, more accelerated decline, decreased vascular co-morbidity and higher tau burden than older subjects, among others.

- Participant 3 had a disease duration of 3 months.

- Please describe the motivation for people to participate in this short-term and invasive phase 1 study

Results

Significant safety concerns are listed below:

- Participant 3 had a 4 point decline on the MMSE and a 5 point change on the ADAS in 2 months. Did that represent an improvement or decline on the ADAS?

- Participant 4 had a 4 point decline on the MMSE in 1 week
- Participant 2 had a 16 point increase on the NPIQ and 15 point decline in the ADCS ADL at 2 months
- Two participants had post-procedural hypointensities on gradient echo

Discussion

- The sentence which begins on line 248 is speculative and should be removed. The anecdotal report of the sentence that begins on line 250 should be removed.
- The dramatic asymmetry and post-procedural change on amyloid PET in participant 5 is fascinating but hard to explain. No conclusions can be drawn from this observation.

Please describe the Focused Ultrasound Foundation that funded the study.

Response to Reviewers' comments:

Reviewer #1 (Remarks to the Author):

Lipsman and colleagues pursue an innovative approach to treat Alzheimer's disease that so far has remained incurable. Their paper reports a small safety study with five MCI patients that had undergone a targeted opening of the blood-brain barrier (BBB) in the non-eloquent cortex in the right frontal lobe using a two-staged approach, with a targeted area that was twice as large in stage 2 compared to that in stage 1. To achieve BBB opening the team of Dr Lipsman and colleagues used MR-guided focused ultrasound. The team reports safety and successful BBB opening as the primary outcome) and a possible reduction in amyloid-beta as measured with [18F]-florbetaben PET at the targeted region in some patients, suggesting an effect on plaque clearance (secondary outcome).

[We thank the reviewer for their time and comments, and share their interest in this novel approach for Alzheimer's Disease.]

Safety: From a safety point of view this proof-of-concept study is promising as proven by a battery of read-outs (radiographic, neuropsychological).

[We appreciate the comments regarding the safety of this procedure, which was the objective of our phase I trial.]

(1) Could the authors expand why they avoided areas containing sulci and vessels. Do they expect an effect of the microbubbles on large vessels?

[As this was a pilot trial, our objective was to demonstrate, for the first time, safe non-invasive BBB opening in amyloid positive brain. It is unclear, at present, what the effects would be on large vessels, similar to the caliber of those in human brain sulci. Accordingly, and to mitigate additional risk, we sought to avoid sulci and associated vessels in this study in the unlikely, but nevertheless possible, chance that this could lead to undue risk for patients. With safety data now obtained, we can more confidently target close to sulci, in subsequent trials. We referred to this in paragraph 3 of the Discussion (page 9, line 288).]

BBB opening: It is good to find that the BBB closes after one day as shown by gadolinium leakage. The authors report about a learning curve and indeed, as Fig S1 shows, BBB opening has not been uniformly achieved.

[Reversal of BBB opening is a key component of our study, and we established that the BBB is closed the day following the procedure. Uniformity of BBB opening is determined by factors such as microbubble dosage, timing of sonication relative to microbubble injection, and volume and type of tissue being opened. We have removed the reference to a learning curve from our manuscript, and instead now define more precisely the factors that can contribute to more effective BBB opening (page 9, line 298). The determination of each cavitation 'threshold' is a process described recently, by our group (Huang, et.al Radiology 2017), and also in response to Reviewer 2's comment ii). We agree with the reviewer, that this will enhance reproducibility of our results, and more clearly explains the methods.]

(2) How did the team space out the sonications? Did they do stage 1 and then 2 on patient 1, and then move to patient 2, then 3, then 4 and 5, or did they do all stage 1 sonications pretty much at the same time?

[The reviewer is correct, however we sought, to the extent possible, to group patients in twos, such that patients 1 and 2 underwent stage 1 on the same day, and then 1-month later, stage 2 on the same day. Patients 3 and 4 were similarly grouped for stage 1 on the same day, but only patient 3 went on to have stage 2. Patient 5 was then treated with stage 1 and then stage 2.]

(3) Lines 133 to 138 require some further information. What do the authors mean by 3-by-3 spot grid -2-by-2 for the last three patients? How many spots in total? Also, what was done with the first two patients? By sonicating each spot in burst mode for 300ms how do they end up being 'precisely 2ms on and 28ms off'?

[Blood brain barrier opening by the natural focus of our ultrasound device is a spot approximately 2.5 to 3mm in diameter. To increase the coverage volume, electronic steering of the ultrasound beam was applied over a 3x3 grid with 3mm spacing (i.e. 9mm by 9mm in area). This was done for the first two patients. Given the significant cerebral atrophy observed in patients with Alzheimer's disease, we made the decision to reduce the volume of BBB opening for the last 3 patients, and use a smaller grid of 2x2 with 2.5 mm spacing (i.e. 5mm by 5mm in area). The sonication at each sub-spot was 300ms in duration with 2ms on/28 ms off pulses, i.e. 2ms on, 28 ms off, 2ms on, 28ms off, and so, for 300ms. We clarified and now include these details in the revised manuscript (page 5, line 161)].

Amyloid:

(4) An 18F-flobetaben PET scan is shown for patient 5 (Fig 3). Why is there a hotspot of PET signal in the right hemisphere? Considering the very small area that has been targeted how do the authors explain the dramatic effect in the entire hemisphere? Is it possible that in Fig 3C less label has been taken up?

[Lateralization of beta-amyloid deposition has been reported in the literature and is correlated with lateralization of neuronal dysfunction (Frings et al 2015). In Patient 5, beta-amyloid uptake at baseline was asymmetric.]

We obtained beta amyloid imaging in our patients to confirm the diagnosis and to ensure amyloid was present at the sonicated region. We have now emphasized in our revised manuscript that exploratory analysis of amyloid imaging did not yield a clear pattern, in either direction, of the effect of FUS BBB opening on amyloid deposition (page 9, 343). For a more comprehensive presentation of our results, we further include the pre and post sonication amyloid PET images of all patients in Figure 3 and expanded images of all patients in the supplementary material. We do not believe that there was less tracer uptake in figure 3C. Radioligand manufacture, quality and dosing was strictly controlled, and comprehensively checked, and the timing of injection and scan judiciously timed.]

(5) How do the scans look like for patients 1-4?

[This is now included in the revised figure 3 and in the supplementary material.]

(6) Looking at Table S2, it seems that for the first two patients the SUVr value even goes up. The values were calculated by normalizing to the cerebellar gray matter. How does the values look like for the corresponding frontal area on the contralateral non-targeted side (again by normalizing for the cerebellar gray matter).

[With FBB ligand there is a +/-6% variability in tracer uptake with repeated scans. Comparing the baseline to treatment 1, the contralateral hemisphere shows increases in the SUVr in the first patient, decreases in patients 2-4 and an increase in patient 5. The increase in patient 5 is likely due to the low SUVr in that hemisphere. Comparing the baseline to treatment 2, the contralateral hemisphere shows increases in the SUVr in all patients, consistent with the treated hemisphere except for patient 5 who showed a decrease after treatment 2. We now include this information in the supplementary material (Table S3).]

Typo line 131/132: 'Further sonications were delivered energy at half the power ...

[This has been corrected in the revised manuscript]

In my view the safety and BBB opening is exciting and convincing, but the reduction in amyloid is based on one patient only (with a surprising asymmetry) and should be removed as a statement from both the abstract and the body of the text.

[We thank the reviewer for their comments, particularly surrounding the safety of BBB opening.]

We have now modified the abstract (page 2, line 42) and mention in the manuscript that exploratory analysis of PET findings yielded no clear signal on the effect of BBB opening on amyloid deposition, in any direction. Figure 3 now includes amyloid PET scans of all five patients, and we re-state in our discussion that conclusions cannot be drawn from our early findings, particularly from group-wise exploratory analysis (page 9, 343).

We agree with the reviewer that it is imperative to interpret PET findings with an abundance of caution. Additional patients are needed, in larger studies. Our aim was to present the data as neutrally and objectively as possible. We have as a result revised the wording in our abstract and body to better reflect this.]

Reviewer #2 (Remarks to the Author):

Comments for Author:

Lipsman et al describe opening the blood brain barrier in patients with Alzheimer's disease using focused ultrasound and microbubbles. They suggest that using their methodology it was possible to demonstrate a transient opening that occurred following treatment, in the blood brain barrier that appeared to have some association with reduction in fibrillary amyloid in a focused area of the brain in 5 subjects with Alzheimer's disease, but due to small numbers and lack of placebo control, it is unclear whether there were any effects on cognition.

The results are interesting from a treatment standpoint as well as documenting potential safety prior to possible further steps towards a therapeutic clinical trial.

[We thank the reviewer for their time and comments on our manuscript. In particular, we appreciate their recognition of our study's documentation of safety]

Comments as follows:

i) Patients selected appear to be relatively young, at least 4 of the 5 were in their 60s; raising the question was there a bias toward accepting younger patients for some reason? More detail regarding why patients were excluded would be helpful.

[Our revised manuscript now includes information about the total number of patients screened, and the reasons for their exclusion (page 4, line 119).

Patients were screened as they were referred to our study for consideration. Referrals came from within our institution and outside of it by community neurologists and primary care physicians. Patients were then assessed by experts in the diagnosis and treatment of dementia, had their diagnosis confirmed clinically, and underwent amyloid scans. Patients were further assessed according to the inclusion/exclusion criteria, without bias or pre-selection. Although most of our patients were in their early 60's, our intention was not to select younger patients, but rather patients exclusively meeting inclusion/exclusion criteria, specifically without bias, given the phase I nature of our trial. We now include this in the limitations section of our paper (page 9, line 370).]

ii) What is the "ramp" test?

A ramp test refers to the incremental process for determining the sonication power for blood-brain barrier opening. This occurs in conjunction with the initial microbubble injection at each target. Briefly, during the initial sonication, energy increasing in 5% power increments is administered until the device hydrophones detect a sub-harmonic signal, which suggests cavitation. Subsequently, 50% of this 'cavitation threshold' power is the energy used for the remaining sonications at that target location. We include this definition and information in our revised manuscript (page 5, line 155). This algorithm was developed based on preclinical data published by our group and further detail is found in the following:

O'Reilly, M. A. & Hynynen, K. Blood-brain barrier: real-time feedback-controlled focused ultrasound disruption by using an acoustic emissions-based controller. Radiology 263, 96–106 (2012)

Huang Y., Alkins R, Schwartz, M.L. et al. Opening the blood-brain barrier with MR imaging-guided focused ultrasound: preclinical testing on a trans-human skull porcine model. Radiology 282, (123-130) 2017.

iii) Several of the patients were on SSRIs that have been suggested to utilize a transporter protein to cross the blood brain barrier. Any comments on effects or interaction of their technique with these drugs.

[The reviewer raises an important point regarding the co-circulation of medical therapies in the context of BBB opening. This is especially important given the potential to ultimately couple BBB opening with the delivery of a therapeutic agent. We did not note any obvious interactions between SSRI's and BBB opening. Depression ratings did not change, and we did not detect any clinically significant changes in neurologic or other function throughout study follow-up. In a subsequent phase II trial, additional measures, including serum and CSF, will be used to characterize in a larger number of patients, the influence of BBB opening on the release and passage of materials into and out of the brain.]

iv) It was stated technical performance improved with every patient and results improved with each following patient. This does not describe study design and conduct in a reproducible way for the reader. What aspects of methodology improved?

[The quality of BBB opening depends on several factors including sonication parameters (e.g. power, frequency, duration, duty cycle), microbubble dosage, volume and type of target tissue, as well as factors that improve with operator experience such as microbubble handling and injection technique, and coordination between microbubble injection and sonications. In our revised manuscript, we more precisely define and discuss the factors that contribute to effective BBB opening, and hopefully this will enhance the clarity and reproducibility of our results. (page 8, line 298)]

v) Florbetapir was used before these studies. With application of technique were the regions treated all previously documented positive for amyloid? Were there any broader effects on reduction in brain amyloid?

[We used Florbetaben in our study, rather than Florbetapir. Both are radioligands for beta-amyloid, but made by different manufacturers. All patients underwent baseline amyloid PET imaging, and required, as per our inclusion criteria, to have amyloid present in the target region. For statistical reasons, given the sample size and volume of region targeted, we restricted our analysis to the sonicated region and exploratory analyses to the contralateral side and other regions of interest. This information is included in the supplementary material. In our revised manuscript, we note in our abstract and body that the PET analysis no clear signal on the influence of BBB opening on amyloid deposition, in either direction. (page 2, line 42, page 9, line 336)]

vi) In summary very interesting data, but report would be improved with more details regarding methodology and study design.

[We again thank the reviewer for their interest in our work. Their comments have strengthened our paper and the discussion of its methods.]

Reviewer #3 (Remarks to the Author):

Lipsman and colleagues studied the safety of open treatment with MR-guided ultrasound in the right frontal lobe with intravenously injected microbubbles performed twice over 4 weeks in 5 patients with mild-moderate Alzheimer's disease. They hypothesized that this technique would alter the blood brain barrier. They used transient local gadolinium enhancement as evidence of change in the blood brain barrier. They state that the procedure was well-tolerated but there were some safety signals.

[We thank the reviewer for their time and comments.]

Abstract-The statement regarding beta amyloid reduction is overstated and should be removed. The statement regarding safety should be revised.

[We have modified the amyloid statements in our abstract and manuscript, as described in sections of Reviewers 1 and 2 above (page 2, line 42, page 9, line 336)]

Methods-Please report the reasons that 16 participants were excluded.

[We have added this information to the manuscript in the methods section (page 4, line 119).]

- 4/5 participants were <65 which is not representative of the age range of individuals with sporadic AD. Testing the safety and efficacy of this technique in early onset AD may not be generalizable to older people with sporadic AD as early onset patients have a higher APOE carriage rate, more accelerated decline, decreased vascular co-morbidity and higher tau burden than older subjects, among others.

[As addressed in response to reviewer 2, our intention was not to specifically enroll patients less than 65, but rather screen and enroll patients as they were referred to our study, in as unbiased and objective way as possible. Patients were screened according to the inclusion/exclusion criteria (age 50-85, etc). We agree with the reviewer that early onset patients, may represent a distinct subgroup. However, as this was a phase I, feasibility study, designed to evaluate safety of non-invasive BBB opening in Alzheimer's Disease, the intention was to enroll patients with confirmed diagnosis, and amyloid positivity in the target region, regardless of genotype status. We agree, and emphasize in our manuscript, that generalizability is limited. We have now included this in our paper as an additional limitation (page 9, line 370). In the design of a subsequent Phase IIa trial, genetic screening for APOE4 status will identify this patient cohort, making generalizability of future findings to a larger AD population possible.]

- Participant 3 had a disease duration of 3 months.

[Patient 3 received a formal diagnosis of Alzheimer's disease 3 months prior to our study, but was symptomatic for a significantly longer period]

- Please describe the motivation for people to participate in this short-term and invasive phase 1 study

[The reviewer raises an important point, and one which applies to all phase I, pilot, safety, feasibility studies. Our group has a large experience with such trials, using various surgical modalities, and especially focused ultrasound. From our experience and in speaking to the five patients enrolled, as well as the 16 who failed formal screening, and the countless others who expressed interest, the motivation for participating in such studies is primarily altruism and being part of an effort to develop a disease-modifying treatment where no other exists. No one is more acutely aware of the urgent need for novel therapeutics than patients and their caregivers. The primary motivation is to advance the field towards new treatments. Other motivating factors include the possibility of participating in future trials and the theoretical benefit from being part of this study (i.e. pre-clinical evidence suggesting amyloid reduction with FUS BBB opening). Critically, in our informed consent discussion, we highlight that there is no expectation of clinical benefit from this study.]

Results

Significant safety concerns are listed below:

- Participant 3 had a 4 point decline on the MMSE and a 5 point change on the ADAS in 2 months. Did that represent an improvement or decline on the ADAS?

[Patient 3's ADAS-cog changed from 30 during screening to 25 at last follow-up. This 5-point change is an improvement on the test. On the other hand, his MMSE declined from 22 to 18 during the same time period. These changes highlight the test-retest variability in cognitive measures, and their inherent limitations in the context of a neurodegenerative illness such as AD, that apply to all therapeutic trials. For instance, Clark et al. found retest-test variability of MMSE for patients in the Consortium to Establish a Registry for Alzheimer's Disease ranged from +4 to -6. In addition, as with the other scores, it is critical to put tests in context of clinically meaningful changes. In no instance during our study did a patient, their primary caregiver, the treating physician(s), or the research team, detect a clinically meaningful change in cognition and/or behavior. As highlighted by the two other reviewers, this provides compelling safety data for focal and reversible non-invasive BBB opening in amyloid positive brain.

Clark, C. M. et al. Variability in Annual Mini-Mental State Examination Score in Patients With Probable Alzheimer Disease: A Clinical Perspective of Data From the Consortium to Establish a Registry for Alzheimer's Disease. Arch. Neurol. 56, 857–862 (1999).]

- Participant 4 had a 4 point decline on the MMSE in 1 week

[As with patient 3, although this represents a decline in MMSE, their ADAS-cog decreased by 4 points as well at this time point, representing an improvement in function. Personal circumstances surrounding the day of testing lead to this participants' self-reported anxiety prior to his MMSE assessment (e.g. his anticipation of 'looking for improvement' during the cognitive assessment itself) and significant early morning travel contributing to fatigue, which is known to influence cognitive scores in AD. Additional family stressors were discussed upon interview, but remain confidential. In our revised manuscript, an explanatory note has been added to the table. As above, we do not believe this raises a safety concern for BBB opening.]

- Participant 2 had a 16 point increase on the NPIQ and 15 point decline in the ADCS ADL at 2 months

[An untimely change in personal circumstances (marital) contributed to a new caregiver reporting for this visit, which would be reflected in the decline in ACDS ADL score. Similarly, as the NPIQ score is a caregiver's view of the patient, any extenuating circumstances creating stress for either the patient or interviewee may be reflected.]

[On post-study follow-up, no adverse changes in function or cognition were reported by the patient or caregiver. Notably, the patient has requested to be screened for enrollment in the proposed next phase of this trial.]

- Two participants had post-procedural hypointensities on gradient echo

[Two patients did have post-procedural changes on gradient echo (GRE). Hypointensity on GRE can be caused by a microbleed in the parenchyma, blood in the perivascular space, or contrast material (gadolinium) in the perivascular space. If blood stains the parenchyma, hypointensities persist for weeks and months, and can be visible on subsequent imaging. In both instances, we did not see GRE changes on follow-up imaging, indicating resolution. This argues against microbleed, and in favour of perivascular leakage, which is consistent with focal BBB opening, and the known mechanism of MRgFUS and microbubbles.]

[The reviewer has raised several points, specifically about their safety concerns, and we appreciate their comments and the opportunity to address these. Safety, as determined by radiographic or clinical adverse events were primary outcome measures in our Phase I trial. Adverse events can be defined as serious or non-serious as well as treatment- or non-treatment related. Serious adverse events are those that require additional treatment or hospitalization(s). Treatment-related adverse events can be attributed directly to the intervention, to the underlying illness, or other factors. Our study saw no serious adverse events; there were no bleeds, swelling, or any medical or other complaints. There were further no clinically significant changes in cognitive measures, or radiographic outcomes. These are critical safety findings, when one considers that the BBB was opened temporarily, in a brain region that was heavily positive for amyloid.]

Discussion

- The sentence which begins on line 248 is speculative and should be removed. The anecdotal report of the sentence that begins on line 250 should be removed.

[These sentences have been removed from the manuscript]

- The dramatic asymmetry and post-procedural change on amyloid PET in participant 5 is fascinating but hard to explain. No conclusions can be drawn from this observation.

[We agree with the reviewer that, though fascinating, we cannot draw conclusions based on the amyloid results in this patient, and have included this statement in our manuscript. We have revised our abstract and body appropriately, and have modified Figure 3 to now include PET images from all 5 patients, as well as expanded images in the supplementary material. We emphasized that our study was not designed to study efficacy, but rather safety (page 2, line 42, page 9, line 336). Further study, in the context of a larger trial, in more patients

is now required, and our trial provides the safety data permitting such studies to now take place.]

Please describe the Focused Ultrasound Foundation that funded the study

[The Focused Ultrasound Foundation (FUSF) is a not-for profit charitable organization that seeks, through funding of investigator and industry sponsored studies, to accelerate research in focused ultrasound, with an aim towards novel indications. Our work was funded following a peer-reviewed evaluation of our application and proposal. InSightec, the device manufacturer and regulatory sponsor, did not fund this study, nor did they have input on study design, methods or analysis.]

Reviewers' comments:

Reviewer #1 (Remarks to the Author):

(1) The authors have toned down the key statement in the Abstract and Discussion that ultrasound removes amyloid; however, the statement is still there in the Results:

'Amyloid reduction at the treated ROI was observed in patients 4 and 5 after stage 1 and in patient 5 after stage 2' (results page 8 lines 225-226).

This claim cannot be made because, as shown in Table S3 (that is not even cross-referenced in the manuscript and should be added to the main text rather than the supplement) the PET SUVR signal varies too much to draw this conclusion. For example patient 5 state 1 shows as much a decrease in the treated hemisphere as there is an increase in the contralateral hemisphere. The explanation in the rebuttal letter (point 6) is not really satisfying and this information is not found in the manuscript text.

(2) Also, no explanation is being provided (point 4) why targeting a tiny brain area would remove amyloid in the entire hemisphere.

(3) The other points have been addressed to my satisfaction.

Reviewer #2 (Remarks to the Author):

The authors have been responsive to my comments and the comments of my fellow reviewers and the report is substantially improved as a result. The continued improvement with experience that were not fully explained in description of methodology are better described but some details remain opaque. No further comments.

Reviewer #3 (Remarks to the Author):

Lipsman and colleagues has submitting a revised version of their pilot safety study on focused ultrasound to treat Alzheimer's disease. They have addressed a number of the reviewers' concerns but significant concerns remain.

- They claim there are no safety signals but patient 2 had a decrease of 4 points on the MMSE at 1 month and a 15 point decline in ADL and 16 point worsening in NPIQ at 2 months, patient 3 had a 4 point decline on MMSE at 2 months, patient 4 had a 4 point decline on MMSE and a 6 point decline on ADL at 1 week, and patient 5 had a 6 point decline on ADL at 2 months. There is concern regarding the reliability of the cognitive assessments.
- Remove the caveats in Table 3 for declines recorded.
- Briefly describe InSightec and the Focused Ultrasound Foundation in the section on Funding Source. These entities will be unknown to readers.
- Line 195 state that 21 were screened and 5 were enrolled and describe the reasons the 16 were excluded in the text or Figure 1.
- Briefly describe the ramp test in the text.
- Revise the paragraph starting with line 224 to state that no significant differences were seen in amyloid PET and direct the reader to supplementary figures 3-8 and Tables S2 and 3. Order the supplementary PET figures by patient number.
- Remove the statement on lines 249-251.
- Remove the material from the end of line 259-265, the material from the end of 275-278 and the sentence beginning on line 283.
- Only 5% of AD cases are under 65. State that the young age of the small group in this study may limit generalizability to the vast majority of older patients with AD and higher levels of cerebrovascular and medical co-morbidity.

Response to Reviewers' comments:

Reviewer 1

(1) The authors have toned down the key statement in the Abstract and Discussion that ultrasound removes amyloid; however, the statement is still there in the Results:

'Amyloid reduction at the treated ROI was observed in patients 4 and 5 after stage 1 and in patient 5 after stage 2' (results page 8 lines 225-226).

This claim cannot be made because, as shown in Table S3 (that is not even cross-referenced in the manuscript and should be added to the main text rather than the supplement) the PET SUVR signal varies too much to draw this conclusion. For example patient 5 state 1 shows as much a decrease in the treated hemisphere as there is an increase in the contralateral hemisphere. The explanation in the rebuttal letter (point 6) is not really satisfying and this information is not found in the manuscript text.

[This line (225-226) has now been removed from the revised manuscript, and we also now reference Table S3 in the manuscript.]

(2) Also, no explanation is being provided (point 4) why targeting a tiny brain area would remove amyloid in the entire hemisphere.

[Our study was not designed to evaluate the efficacy of focused ultrasound on amyloid clearance, but rather on the safety of non-invasive BBB opening in amyloid positive AD. Pending larger trials in more patients, designed to study this question specifically, we cannot comment or speculate, beyond theorizing, on the effects, locally or more remotely, on amyloid clearance.]

In accordance with the reviewers' comments, we have removed reference to plaque reduction in any patients, and provide the SUVR and imaging data in supplementary material for the reader's review. Though we demonstrate clinical and radiographic safety, we cannot draw conclusions regarding amyloid removal post-sonication, within or outside of the treated volume. Our acquired data, however, does demonstrate for the first time safe and reversible non-invasive BBB opening in amyloid rich areas, now permitting future trials, in larger cohorts to focus on characterizing the movement of amyloid across the BBB.]

(3) The other points have been addressed to my satisfaction.

[We thank the reviewer for their time and comments.]

Reviewer 2

The authors have been responsive to my comments and the comments of my fellow reviewers and the report is substantially improved as a result. The continued improvement with experience that were not fully explained in description of methodology are better described but some details remain opaque. No further comments.

[We thank the reviewer for their time. We are pleased they found our revised manuscript satisfies both their and the other reviewers' comments.]

Reviewer 3

Lipsman and colleagues has submitting a revised version of their pilot safety study on focused ultrasound to treat Alzheimer's disease. They have addressed a number of the reviewers' concerns but significant concerns remain.

- They claim there are no safety signals but patient 2 had a decrease of 4 points on the MMSE at 1 month and a 15 point decline in ADL and 16 point worsening in NPIQ at 2 months, patient 3 had a 4 point decline on MMSE at 2 months, patient 4 had a 4 point decline on MMSE and a 6 point decline on ADL at 1 week, and patient 5 had a 6 point decline on ADL at 2 months. There is concern regarding the reliability of the cognitive assessments.

[Thank you again for the feedback and comments. Regarding patient 2, 3, and 4's MMSE changes, we saw no significant changes in the corresponding ADAS-cog, which a previous study by the Alzheimer's Disease Neuroimaging Initiative (ADNI) suggests may be a more precise measure of cognitive function (Balsis et al., 2015). These patients' scores at the end of study remained stable compared to baseline, and for some patients actually improved (e.g. patient 3 and 4 from 30 and 19, to 25 and 15, respectively).

The variability seen in these measures reflects their inherent, and established, variability, and are difficult to disentangle from the natural progression of the patient's disease. We agree with the reviewer that the reliability of any specific cognitive assessment is not ideal, and hence multiple tests are often administered, as in our trial. Safety should have a more robust and congruent effect on all tests.

The ADL and NPIQ take into account the patient's environment and their caregiver, and thus contain an even higher degree of variability. The changes noted for patient 3, 4, and 5, are relatively small (max ADL score is 78, max NPI scores is 180) and are within what is expected for progression in a complex neurodegenerative disease. For patient 2, the 16-point increase reflects at least in part a change in the caregiver who responded to the questionnaire. Finally, in designing this phase I study,

the primary outcome was determined a priori to be expert clinical examination and radiologic safety, which were both satisfied.

Our work and data were reviewed by our Research Ethics Board, and subject to review by Health Canada and a Data Safety Monitoring Board. None of these independent bodies identified a safety concern. Indeed, all treated patients have expressed interest in enrolling in the subsequent trial, an indirect measure of the study's tolerability for both patients and their caregivers.]

- Remove the caveats in Table 3 for declines recorded.

[This has been removed from the revised manuscript.]

- Briefly describe InSightec and the Focused Ultrasound Foundation in the section on Funding Source. These entities will be unknown to readers.

[We have added the following to the Funding Source section of our manuscript:

“InSightec, the manufacturer of the ExAblate device used in this study, was the regulatory sponsor and had no role in study design, data collection, analysis or interpretation. This study was funded by a grant from the Focused Ultrasound Foundation, a non-profit organization that funds research into clinical applications of ultrasound.]

Line 195 state that 21 were screened and 5 were enrolled and describe the reasons the 16 were excluded in the text or Figure 1.

[Figure 1 in our revised manuscript now includes the specific reasons for exclusion of these patients.]

- Briefly describe the ramp test in the text.

[A description of the ramp test and its purpose is now expanded on lines 133-138. The purpose of the ramp test is to determine the optimal acoustic power for BBB opening. The system is programmed to apply short ultrasound pulses with increase power at 5% increments until the hydrophones in the

ultrasound device detect subharmonic signal on acoustic feedback. BBB disruption is optimally and safely achieved at 50% of this power.]

- Revise the paragraph starting with line 224 to state that no significant differences were seen in amyloid PET and direct the reader to supplementary figures 3-8 and Tables S2 and 3. Order the supplementary PET figures by patient number.

[The lack of statistical significance is now explicitly stated (line 235). Supplementary PET figures are reordered by patient number.]

- Remove the statement on lines 249-251.

- Remove the material from the end of line 259-265, the material from the end of 275-278 and the sentence beginning on line 283.

[We have removed these statements from our revised manuscript.]

- Only 5% of AD cases are under 65. State that the young age of the small group in this study may limit generalizability to the vast majority of older patients with AD and higher levels of cerebrovascular and medical co-morbidity.

[Our revised manuscript now reflects this limitation on line 290-292.]

REVIEWERS' COMMENTS:

Reviewer #3 (Remarks to the Author):

The authors have satisfactorily responded to the reviewers' concerns.

Response to Reviewers' comments:

Reviewer #3:

The authors have satisfactorily responded to the reviewers' concerns.

[We thank the reviewer for their time, and are pleased that their concerns have been addressed.]